# Efficacy and safety of eszopiclone combined with drug therapy in the treatment of insomnia after stroke: A network meta-analysis and systematic review

**Ruo-Yang Li** [1]*, **De-Liang Zhu**[1], **Ke-Yu Chen**[2]

**1** Geriatric Diseases Institute of Chengdu, Department of Rehabilitation, Chengdu Fifth People's Hospital (The Second Clinical Medical College, Affiliated Fifth People's Hospital of Chengdu University of Traditional Chinese Medicine), Chengdu, China, **2** Department of traditional Chinese medicine, Chengdu Second People's Hospital, Chengdu, Sichuan, China

* alis7718@outlook.com

**Data Availability Statement:** All relevant data are within the paper and its Supporting Information files.

## Abstract

### Objective

To evaluate the efficacy and safety of multi-drug therapy based on eszopiclone in the treatment of insomnia after stroke using a network meta-analysis method and to provide evidence for clinical practice.

### Method

Computer searches of PubMed, Excerpt Medica Database (Embase), Cochrane Library Central Register of Controlled Trials, APA PsycInfo, CNKI, WanFang, Sinomed and other databases were performed to search for clinical randomized controlled studies (RCTs) on multi-drug therapy based on eszopiclone in the treatment of insomnia patients after stroke. The search time was from the establishment of each database until July 2023. The bias risk assessment tool recommended by Cochrane was used to evaluate the quality of the included RCTs. Stata 14.0 was applied to perform network meta-analysis using Review Manager 5.3 software for traditional meta-analysis.

### Result

Eighteen RCTs and 1646 patients were ultimately included, involving 11 treatment options. The results of the network meta-analysis showed that the ranking of Pittsburgh Sleep Quality Index (PSQI) decline was eszopiclone combined with sweet dream oral liquid (ESZ+SDOL)>eszopiclone combined with a shugan jieyu capsule (ESZ+SGJYC)>eszopiclone combined with agomelatine (ESZ+AGO)>eszopiclone combined with flupentixol and melitracen tablets (ESZ+FMT)>eszopiclone combined with yangxue qingnao granules (ESZ+YXQNG)>eszopiclone combined with mirtazapine (ESZ+MIR)>ESZ>FMT; the modified Edinburgh Scandinavia Stroke Scale (MESSS) decline ranking was ESZ+SDOL>ESZ+AGO>ESZ; and the clinical total effective rate ranking was eszopiclone combined with a

**Funding:** The author(s) received no specific funding for this work.

**Competing interests:** The authors have declared that no competing interests exist.

xuefu zhuyu capsule (ESZ+XFZYC)>ESZ+MIR>ESZ+SGJYC>ESZ+SDOL> ESZ +FMT>ESZ+YXQNG>ESZ>FMT. In terms of clinical adverse reactions, in addition to ESZ therapy, ESZ+ESC had the highest number of adverse reactions, with abdominal pain being the most common. ESZ+YXQNG had the most types of adverse reactions, with 8 types.

## Conclusion

Multi-drug therapy based on eszopiclone can effectively improve the sleep quality of patients with insomnia after stroke, and ESZ+SDOL has significant efficacy and safety. However, due to the limitations of this study, efficacy ranking cannot fully explain the superiority or inferiority of clinical efficacy. In the future, more multicentre, large sample, double-blind randomized controlled trials are needed to supplement and demonstrate the results of this study.

## Introduction

Stroke, one of the most common diseases in the world, is characterized by a high incidence rate, recurrence rate, disability rate and mortality. The motor dysfunction, cognitive dysfunction, swallowing dysfunction, central nervous system pain, sleep disorders and other sequelae caused by its onset seriously burden the lives of patients and their families. Insomnia after stroke is one of the common sequelae of stroke, and relevant research data show that more than half of acute stroke patients have sleep disorders such as insomnia [1]. Insomnia after stroke is usually related to environmental factors, psychological factors, stroke site, etc. It has been reported that patients with pontine stroke experience almost complete sleep loss, while stroke in the thalamus leads to a lack of brain waves, leading to insomnia. Stroke in the supratentorial, left hemisphere, or paramedian thalamus can lead to reduced nonrapid eye movements, while stroke in the right hemisphere can lead to reduced rapid eye movements [1, 2]. At the same time, studies have shown that long-term low-quality sleep will increase the risk of relapse and the incidence of poststroke depression [3].

Currently, benzodiazepines, non-benzodiazepines, and antidepressants are commonly used in clinical practice to treat insomnia after stroke. Some scholars believe that benzodiazepines are not recommended for poststroke patients because they may exacerbate respiratory-related sleep disorders and lead to the recurrence of motor deficits [4]. Eszopiclone (ESZ) is a new type of nonbenzodiazepine sedative drug, and its hypnotic mechanism is currently believed to be caused by a Gamma-Amino Butyric Acid receptor complex coupled with benzodiazepine receptors. Its short peak time and active ingredient half-life enable it to maintain its original therapeutic effect even when taken at lower doses. A network meta-analysis of the Lancet suggests that in the treatment of insomnia, ESZ has better clinical efficacy than other commonly used insomnia drugs, but it is also accompanied by a certain incidence of adverse events [5]. To ensure its safety and effectiveness, multi-drug therapy, such as mirtazapine, Flupentixol and Melitracen Tablets and related traditional Chinese patent medicines and simple preparations, with ESZ as the main drug is often used clinically. Its clinical efficacy has also been affirmed to some extent. Currently, there have been many related meta-analysis studies on stroke sequelae, such as poststroke neuralgia and poststroke cognitive impairment [6–8], and published meta-analyses on insomnia after stroke mostly focus on the efficacy observation between single drugs, and few meta-analyses compare the efficacy of drug combination

therapy, especially the combination therapy based on ESZ. This study will be based on a network meta-analysis method using the Pittsburgh Sleep Quality Index (PSQI), modified Edinburgh Scandinavia Stroke Scale (MESSS), clinical total response rate, and clinical adverse reactions as outcome indicators to evaluate the effectiveness and safety of multi-drug therapy based on ESZ in treating insomnia after stroke. Overall, the research results can provide some reference for clinicians.

## Method

### Registration

This meta-analysis is registered on international prospective register of systematic reviews (registration number CRD42023451889).

### Literature search

The search strategy uses a combination of theme words and keywords, and the search terms were adjusted based on the search results. The search time was from the establishment of the database to Oct 2023. The retrieval databases included the following: PubMed, Excerpt Medica Database (Embase), Cochrane Library Central Register of Controlled Trials, American Psychological Association (APA) PsycInfo, China National Knowledge Infrastructure (CNKI), Wanfang Database, Chinese biomedical literature service system. Collect relevant literature on the treatment of insomnia after stroke using multi-drug therapy based on ESZ, search using keywords and their synonyms, and adjust accordingly according to different databases. The search terms mainly include: stroke, insomnia, nonbenzodiazepine, dexzopiclone, eszopiclone, randomized controlled trials, RCTs, et al. The detailed search strategy can be found in the S1 appendix.

### Study selection

1. The research type was RCTs, regardless of whether allocation concealment or blinding methods were used. 2. The study subjects were diagnosed with stroke through imaging examinations such as brain CT and MRI. 3. Stroke patients with insomnia at baseline were included. 4. The treatment plan was as follows. The control group received simple intervention therapy with ESZ, while the treatment group received intervention therapy with multi-drug therapy based on ESZ. 5. The outcome measures were as follows: Main measures: PSQI and MESSS; Secondary indicators: Clinical total effective rate (total effective rate = [(cured + significantly effective + effective) cases ÷ total cases] × 100%) and clinical adverse reactions.

### Study exclusion

The exclusion criteria are as follows (The meta-analysis of comprehensive high-quality randomized controlled trials has been regarded as the highest level of evidence in evidence-based medicine. Therefore, this study only includes RCTs and excludes all other types of studies, as there are not too few included studies.): 1. Overview, cohort studies, animal experiments, case studies, basic research, cross-sectional studies, case reports, etc.; 2. Studies with outcome indicators that did not meet the inclusion criteria; 3. Unable to obtain full text literature; 4. Literature where information could not be extracted, complete original data that were not provided, and data requests that were unsuccessful; and 5. Repeated publications.

## Data extraction

Two trained researchers screened the literature separately. The initial screening was mainly based on the title and abstract. After the initial screening, we searched for literature that met the inclusion criteria and read the full text. If necessary, the original author was contacted to avoid data omission. If the evaluation results of the literature were inconsistent, a third party was invited to participate in the discussion and resolve it. The use of Endnote and Excel software for literature management and data extraction mainly included the basic characteristics, intervention measures, treatment courses, and outcome indicators of the included literature cases. All continuous data were included in the difference in changes before and after treatment (i.e., the difference in indicators after treatment and before treatment). If it was not calculated in the original text, it was calculated by the participant. The formula is as follows [9], corr is usually 0.5. Raw data can be found in the S1 Data.

$$SD_{E\ change} = \sqrt{SD^2_{E\ baseline} + SD^2_{E\ final} - (2 \times Corr \times SD_{E\ baseline} \times SD_{E\ final})}$$

$$Mean_{E\ change} = Mean_{E\ final} - Mean_{E\ baseline}$$

## Risk assessment

According to the bias risk assessment tool recommended in Cochrane Handbook 5.1.0 [10], the included studies were evaluated, including seven aspects: random sequence generation in the literature, allocation concealment, implementation of blinding, whether blinding was implemented for outcome evaluation, completeness of outcome data, whether results were selectively reported, and whether there were other biases. RevMan5.3 software (the Cochrane Collaboration, Nordic Cochrane Center, Copenhagen, Denmark) was used to draw a literature quality evaluation chart.

## Statistical analysis

The network meta-analysis is mainly based on the frequency method, and the specific commands are detailed in the S2 appendix, and it was conducted using STATA 14.0 software (Stata Corporation, Lakeway, TX, USA), and the continuity index (PSQI, MESSS) used the mean difference (MD) as the effect measure. For the binary variable indicator (clinical total effective rate), the odds ratio (OR) was used as the effect quantity, and the corresponding 95% confidence interval (CI) was calculated. STATA was used to draw network evidence relationship maps, forest maps, hierarchical probability maps, funnel maps, and corresponding statistics. When testing global consistency, if the difference is not statistically significant (P>0.05), it indicates that there is no overall inconsistency [11]. Review Manager 5.3 software (the Cochrane Collaboration, Nordic Cochrane Center, Copenhagen, Denmark) was used to conduct subgroup traditional meta-analysis on the main indicators, and heterogeneity testing was mainly determined by I2. If there was no heterogeneity between the research results (I2≤50%), a fixed effects model was used for the meta-analysis; if there was heterogeneity (I2>50%) among the research results, further analysis of the source of heterogeneity was conducted. This study used surface under the cumulative ranking (SUCRA) to calculate the cumulative ranking probability of each treatment plan. The larger the value of SUCRA is, the larger the area under the curve of the cumulative probability ranking chart, indicating a better effectiveness of the intervention measure.

# Results

## Literature search results

According to the literature search results, a total of 211 articles were obtained. After initial screening, 111 articles were excluded, leaving the remaining 100 articles. After reading the title and abstract, 50 articles were excluded, and after reading the entire text, 32 articles were excluded. The total number of articles included in this study was 18 [12–29]. A total of 11 drug therapies were involved, including the following: eszopiclone combined with sweet dream oral liquid (A traditional Chinese patent medicines and simple preparations composed of acanthopanax senticosus, polygonatum, etc. It mainly treats insomnia and dreaminess) (ESZ+SDOL), eszopiclone combined with a shugan jieyu capsule (A traditional Chinese patent medicines and simple preparations composed of Hypericum perforatum L, Acanthopanax senticosus and other traditional Chinese medicines, which mainly treats insomnia and depression.) (ESZ+SGJYC), eszopiclone combined with agomelatine (ESZ+AGO), eszopiclone combined with flupentixol and melitracen tablets (ESZ+FMT), eszopiclone combined with yangxue qingnao granules (A traditional Chinese patent medicines and simple preparations composed of angelica sinensis, rhizoma chuanxiong and other traditional Chinese medicines, mainly used to promote blood circulation and dredge collaterals.) (ESZ+YXQNG), eszopiclone combined with mirtazapine(ESZ+MIR), ESZ, FMT, eszopiclone combined with a xuefu zhuyu capsule (A traditional Chinese patent medicines and simple preparations composed of peach kernel, safflower and other traditional Chinese medicines, mainly used for promoting blood circulation and removing blood stasis.) (ESZ+XFZYC), and escitalopram (ESC), eszopiclone combined with escitalopram (ESZ+ESC). The process and results of literature retrieval are shown in Fig 1.

## Characteristics of the literature

The results of the 18 included studies [12–29] were published from 2014 to 2022, with a total of 1646 patients, including 789 in the experimental group and 857 in the control group. The treatment groups included in the study were all treated with drug combination intervention therapy based on eszopiclone, including 5 studies combined with YXQNG [13, 14, 22, 26, 29], 4 studies combined with SGJYC [15, 18, 19, 24], 3 studies combined with FMT [21, 23, 27], 2 studies combined with SDOL [16, 25], 1 study combined with MIR [12], 1 study combined with AGO [17], 1 study combined with ESC [20], and 1 study combined with XFZYC [28]. In terms of clinical efficacy indicators, 10 studies used clinical efficacy as the outcome indicator [12–14, 16, 22–25, 28, 29], 15 studies used the PSQI score as the outcome indicator [12–19, 21–27], and 3 studies used the MESSS as the outcome indicator [16, 17, 25]. Nine studies reported adverse reactions [15, 16, 18, 20, 22–24, 26, 29]. The basic information of the included literature is shown in Table 1, S1 Table.

## Risk assessment

The included studies were evaluated using the Cochrane Handbook 5.1.0 Bias Risk Assessment tool. In terms of random allocation methods, 11 studies were low-risk and randomly assigned using the random number table method. The remaining studies only mentioned randomness and did not report specific random allocation methods, all of which were medium risk. In terms of the hidden aspects of the random allocation scheme, all included studies were not mentioned and were considered medium risk. In terms of the blinding method for random allocation schemes, none of the included literature mentioned it, so it was considered medium risk. The measurement of outcome indicators included in the literature was not mentioned,

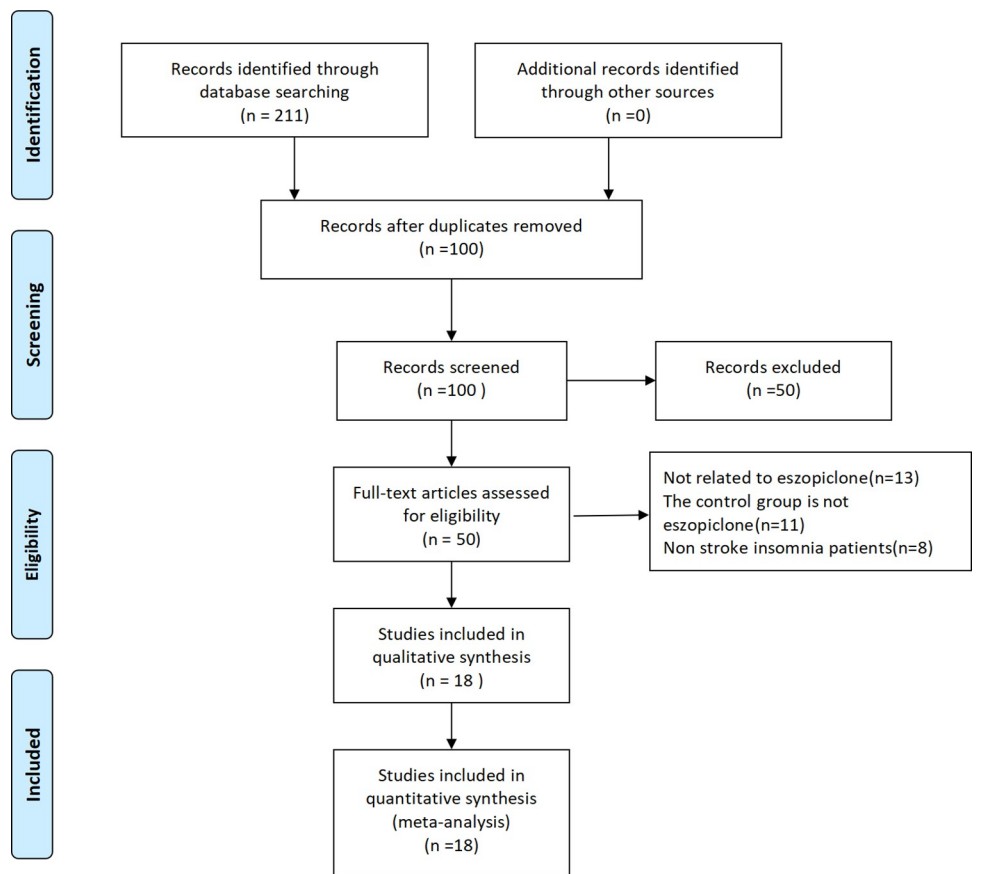

**Fig 1. The process and results of literature retrieval.**

and all were at medium risk. In terms of data integrity, all included research data were complete and low risk. In terms of selective reporting of research results, there was no clear mention of selective reporting in the text, and all included studies were low-risk. Among other sources of bias, there was a high risk [21], as the length of intervention time was not mentioned, while others were all low risk. The risk assessment of bias caused by inclusion in the study is shown in Fig 2.

## PSQI

**Evidence network.**    Fifteen studies reported the PSQI, involving 8 treatment options. The size of the dots represents the sample size for using the intervention measure, and the thickness of the lines represents the number of RCTs using two treatment plans, with a closed loop formed. The number of studies comparing the treatment of ESZ with ESZ+SGJYG and ESZ with ESZ+YXQNG was the highest (4 RCTs). The evidence network of VAS is shown in Fig 3A.

**Inconsistency.**    The 8 treatment options for PSQI forms a triangular ring. The overall inconsistency test results showed P = 0.3329>0.05, indicating that there was no overall inconsistency. The inconsistency factor (IF) was 1.04 (ESZ-ESZ+FMT-FMT). Their IF 95% CI reached zero, indicating no statistical inconsistency, as seen in Fig 3B.

**Network meta-analysis.**    Meta-analysis was conducted on 8 treatment plans, and 16 comparisons showed significant differences. The decrease in PSQI of ESZ+MIR (MD = -2.04, 95%

**Table 1. Basic characteristics of trials included.**

| Author | Year | Number of patients I | Number of patients C | Age(year) I | Age(year) C | Male/female | Treatment I | Treatment C | Intervention period | Outcome indicator |
|---|---|---|---|---|---|---|---|---|---|---|
| Cheng GH [12] | 2022 | 43 | 43 | 48.6±1.32 | 48.51±1.35 | 45/41 | ESZ+MIR | ESZ | 8W | PSQI, Clinical total effective rate |
| Guo YZ [13] | 2017 | 62 | 62 | 56.2±4.2 | 56.3±4.3 | 56/68 | ESZ+YXQNG | ESZ | 4W | PSQI, Clinical total effective rate |
| He M [14] | 2018 | 60 | 60 | 63.3±5.6 | 64.7±4.5 | 59/61 | ESZ+YXQNG | ESZ | 6W | PSQI, Clinical total effective rate |
| Hou CY [15] | 2020 | 43 | 43 | 60.6±13.7 | 60.4±13.6 | 50/36 | ESZ+SGJYC | ESZ | 3W | PSQI, adverse reactions |
| Huang WZ [16] | 2019 | 34 | 34 | 57.82±11.62 | 54.67±10.83 | 37/31 | ESZ+SDOL | ESZ | 4W | PSQI, Clinical total effective rate, adverse reactions, MESSS |
| Li HS [17] | 2019 | 48 | 48 | 62.43±11.12 | 61.82±10.88 | 54/42 | ESZ+AGO | ESZ | 4W | PSQI, MESSS |
| Lu F [18] | 2021 | 41 | 42 | 59.43±8.67 | 60.89±9.02 | 49/34 | ESZ+SGJYC | ESZ | 4W | PSQI, adverse reactions |
| Lv X [19] | 2016 | 41 | 41 | 62.53±11.28 | 62.88±11.46 | 45/37 | ESZ+SGJYC | ESZ | 3W | PSQI |
| Song YM [20] | 2014 | 30 | 30 / 30 | 61±16 | 63 | 27/63 | ESZ+ESC | ESC / ESZ | 4W | adverse reactions |
| Wang SH [21] | 2019 | 20 | 20 | 56 | | 18/22 | ESZ+FMT | ESZ | - | PSQI |
| Wang Y [22] | 2019 | 47 | 45 | 64.8±4.9 | 63.5±5.4 | 45/47 | ESZ+YXQNG | ESZ | 2W | PSQI, Clinical total effective rate, adverse reactions |
| Wang YF [23] | 2021 | 38 | 38 / 38 | 56.94±6.13 | 56.51±6.42 / 57.32±6.35 | 63/51 | ESZ | FMT / ESZ+FMT | 8W | PSQI, Clinical total effective rate, adverse reactions |
| Wei QZ [24] | 2022 | 42 | 47 | 58.59±3.37 | 58.63±3.41 | 53/36 | ESZ+SGJYC | ESZ | 6W | PSQI, Clinical total effective rate, adverse reactions |
| Xiao DF [25] | 2016 | 48 | 48 | 63.52±5.35 | 63.55±5.37 | 51/45 | ESZ+SDOL | ESZ | 4W | PSQI, Clinical total effective rate, MESSS |
| Yang M [26] | 2022 | 43 | 43 | 56.88±10.29 | 56.63±10.16 | 48/38 | ESZ+YXQNG | ESZ | 2W | PSQI, adverse reactions |
| Zhang KM [27] | 2020 | 38 | 36 | 58.34±1.12 | 57.51±2.14 | 44/30 | ESZ+FMT | ESZ | 2W | PSQI |
| Zheng YH [28] | 2022 | 51 | 51 | 62.33±7.64 | 62.49±7.31 | 59/43 | ESZ+XFZYC | ESZ | 8W | Clinical total effective rate |
| Zhu ZQ [29] | 2018 | 60 | 58 | 65.43±7.65 | 65.5±7.66 | 67/51 | ESZ+YXQNG | ESZ | 4W | Clinical total effective rate, adverse reactions |

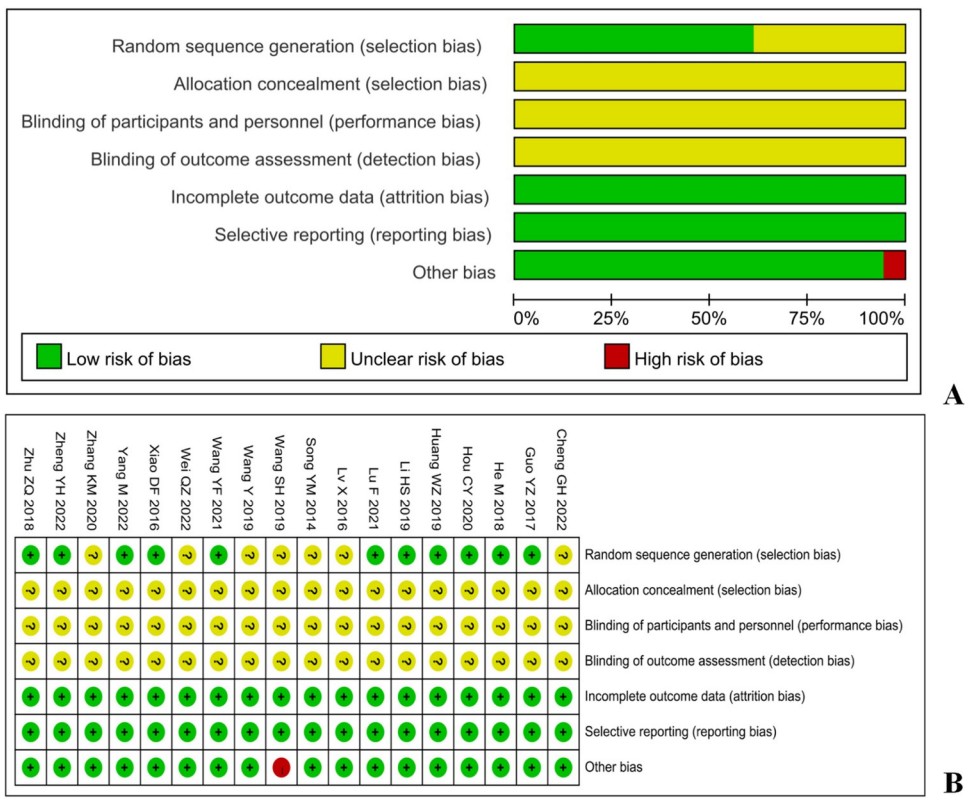

**Fig 2. A: Risk of bias graph; B: Risk of bias summary.**

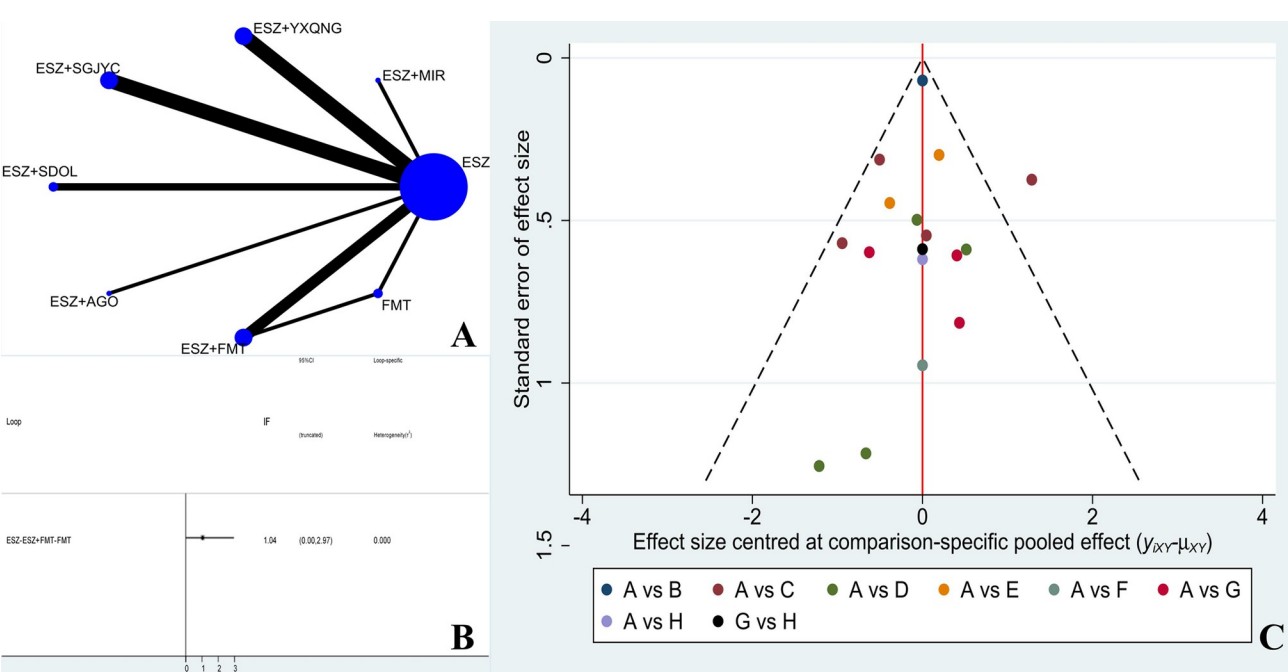

**Fig 3.** A: Network diagram of PSQI; B: PSQI inconsistency testing; C: Publication bias graph of PSQI (A: ESZ; B: ESZ+MIR; C: ESZ+YXQNG; D: ESZ +SGJYC; E: ESZ+SDOL; F: ESZ+AGO; G: ESZ+FMT; H: FMT).

**Table 2. Network meta-analysis of PSQI.**

| Treatment | FMT | ESZ+FMT | ESZ+AGO | ESZ+SDOL | ESZ+SGJYC | ESZ+YXQNG | ESZ+MIR | ESZ |
|---|---|---|---|---|---|---|---|---|
| FMT | - | | | | | | | |
| ESZ+FMT | 3.37 (1.84,4.90)* | - | | | | | | |
| ESZ+AGO | 3.56 (0.86,6.25)* | 0.19 (-2.24,2.61) | - | | | | | |
| ESZ+SDOL | 5.75 (3.91,7.58)* | 2.38 (0.96,3.79)* | 2.19 (-0.22,4.60) | - | | | | |
| ESZ+SGJYC | 3.50 (1.67,5.33)* | 0.13 (-1.28,1.54) | -0.06 (-2.46,2.35) | -2.25 (-3.63,-0.86)* | - | | | |
| ESZ+YXQNG | 3.11 (1.40,4.83)* | -0.26 (-1.52,1.00) | -0.44 (-2.77,1.88) | -2.64 (-3.87,-1.41)* | -0.39 (-1.61,0.83) | - | | |
| ESZ+MIR | 2.32 (0.36,4.27)* | -1.05 (-2.62,0.52) | -1.24 (-3.74,1.26) | -3.43 (-4.98,-1.88)* | -1.18 (-2.73,0.36) | -0.80 (-2.20,0.61) | - | |
| ESZ | 0.28 (-1.27,1.82) | -3.09 (-4.11,-2.08)* | -3.28 (-5.48,-1.08)* | -5.47 (-6.46,-4.49)* | -3.22 (-4.20,-2.25)* | -2.84 (-3.57,-2.10)* | -2.04 (-3.24,-0.84)* | - |

CI = -3.24to -0.84), ESZ+YXONG (MD = -2.84, 95% CI = -3.57 to -2.1), ESZ+SGJYC (MD = -3.22, 95% CI = -4.2 to -2.25), ESZ+SDOL (MD = -5.47, 95% CI = -6.46 to -4.49), ESZ+AGO (MD = -3.28, 95% CI = -5.48 to -1.08), and ESZ+FMT (MD = -3.09, 95% CI = -4.11 to -2.08) was significantly higher than that of ESZ; Compared with ESZ+MIR, the decrease in PSQI of ESZ+SDOL (MD = -3.43, 95% CI = -4.98 to -1.88) was more than that of ESZ+MIR, while the decrease in PSQI of FMT was inferior to that of ESZ+MIR (MD = 2.32, 95% CI = 0.36 to 4.27); Compared with ESZ+YXONG, the decrease in PSQI of ESZ+SDOL (MD = -2.64, 95% CI = -3.87 to -1.41) was more than that of ESZ+YXONG, while the decrease in PSQI of FMT was inferior to that of ESZ+YXONG (MD = 3.11, 95% CI = 1.40 to 4.83); Compared with ESZ+SGJYC, the decrease in PSQI of ESZ+SDOL was more than that of ESZ+SGJYC (MD = -2.25, 95% CI = -3.63 to -0.86), while the decrease in PSQI of FMT was inferior to that of ESZ+SGJYC (MD = 3.5, 95% CI = 1.67 to 5.33); Compared with ESZ+SDOL, the decrease in PSQI of ESZ+FMT (MD = 2.38, 95% CI = 0.96 to 3.79) and FMT (MD = 5.75, 95% CI = 3.91 to 7.58) was less than that of ESZ+SDOL; Compared with ESZ+AGO, the decrease in PSQI of FMT was less than that of ESZ+AGO (MD = 3.56, 95% CI = 0.86 to 6.25); The decrease in PSQI of FMT was inferior to that of ESZ+FMT (MD = 3.37, 95% CI = 1.84 to 4.9), and there was no significant difference in other comparisons, as shown in Table 2.

**Heterogeneity testing and traditional meta-analysis.** Subgroup traditional meta-analysis was conducted on RCTs with more than 2 identical intervention measures within the group. The results showed that there was significant heterogeneity ($I2>50\%$) in the four RCTs of ESZ+YXQNG in the treatment group. After sensitivity analysis and exclusion, the heterogeneity of the results was 0 after excluding one RCT [13]. After careful comparison of the relevant basic information of these four studies, it was found that the baseline PSQI before treatment in the excluded study was lower than that in the other three studies, which may be the fundamental reason for its heterogeneity. The rest of the studies were basically homogeneous ($I2<50\%$), using a fixed effects model. Among the four treatment regimens included, the decrease in PSQI of all combined drug treatment regimens for insomnia after stroke was more significant than that of the simple ESZ group ($P<0. 05$) 05), see Table 3.

**Table 3. Traditional meta-analysis of PSQI.**

| Tteatment | Numbers of RCTs | MD [95%CI] | $I^2$/% | Z | P | Effect |
|---|---|---|---|---|---|---|
| ESZ+SGJYC VS ESZ | 4 | -3.31 [-4.00, -2.63] | 0 | 9.51 | <0.00001 | Fixed |
| ESZ+YXQNG VS ESZ | 4 | -2.81 [-3.83, -1.80] | 82 | 17.08 | <0.00001 | Random |
| ESZ+YXQNG VS ESZ | 3(After removing the RCT with the highest heterogeneity) | -2.34 [-2.82, -1.86] | 0 | 9.55 | <0.00001 | Fixed |
| ESZ+FMT VS ESZ | 3 | -3.07 [-3.81, -2.33] | 0 | 8.12 | <0.00001 | Fixed |
| ESZ+SDOL VS ESZ | 2 | -5.55 [-6.04, -5.06] | 14 | 22.34 | <0.00001 | Fixed |

**Publication bias.**   In this study, eight different treatment regimens were involved in the outcome indicators of the PSQI. The dots of different colours in the funnel plot represent a direct comparison between two different rehabilitation treatment regimens, and the number of dots represents the number of studies. Most of the circles in the funnel plot of this study are symmetrically distributed on the vertical line and its two sides, with basic symmetry on both sides. However, there may still be a certain degree of publication bias. The funnel plot is shown in Fig 3C.

## MESSS

**Evidence network and inconsistency.**   Three studies have reported on MESSS, involving three treatment options. There is no closed-loop formation, so inconsistency testing is not needed. The evidence network of MESSS is shown in Fig 4A.

**Network meta-analysis.**   Three treatment regimens were used for the network meta-analysis of MESSS, and the three comparisons showed significant differences. Compared with ESZ, the decrease in MESSS of ESZ+SDOL (MD = -6.56, 95% CI = -8.95 to -4.17) and ESZ+AGO (MD = -4.38, 95% CI = -7.28 to -1.48) was more significant than that of ESZ, and there was no significant difference in the remaining comparisons, as shown in Fig 4B.

## Clinical total effective rate

**Evidence network and inconsistency testing.**   Eight studies reported overall clinical efficacy, involving eight treatment options. The only closed loop in the network diagram was the direct comparison of the three arms of the same RCT, without forming an indirect closed loop, so there was no need for inconsistency testing. The evidence network for clinical total effectiveness is shown in Fig 5.

**Network meta-analysis.**   Eight treatment plans were subjected to network meta-analysis, and 10 comparisons showed significant differences. Compared with ESZ, the clinical total effective rates of ESZ+MIR (OR = 5.43, 95% CI = 1.1 to 26.83), ESZ+YXONG (OR = 4.05, 95% CI = 2.36 to 6.95), ESZ+SDOL (OR = 4.76, 95% CI = 1.51 to 14.95), ESZ+SGJYC (OR = 5.41, 95% CI = 1.11 to 26.31), ESZ+FMT (OR = 4.17, 95% CI = 1.05 to 16.61), and ESZ+XFZYC (OR = 5.98, 95% CI = 1.24 to 28.83) were significantly higher than those of ESZ. Compared

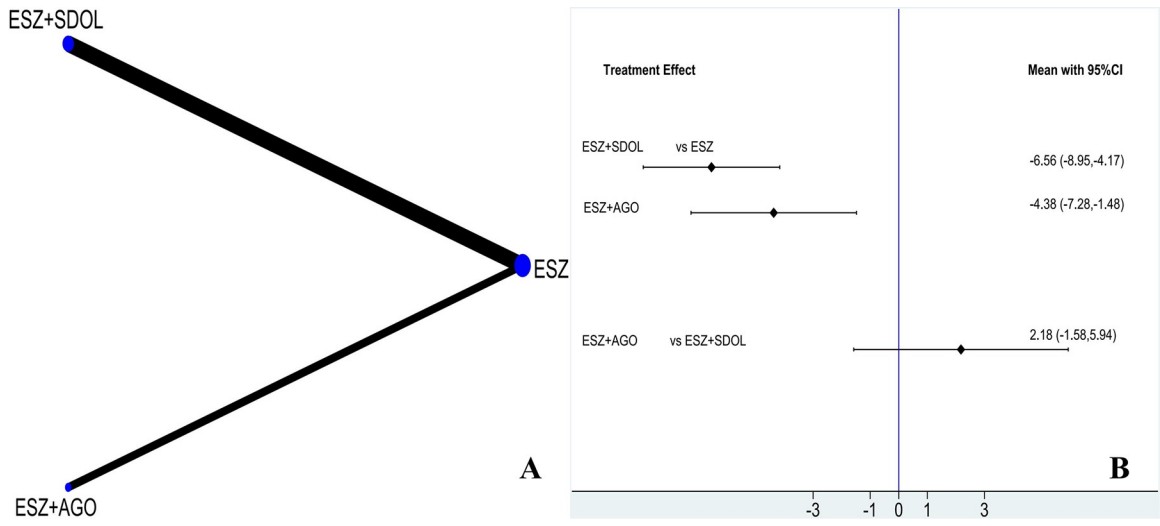

**Fig 4.**  A: Network diagram of MESSS; B: Forest Map of MESSS.

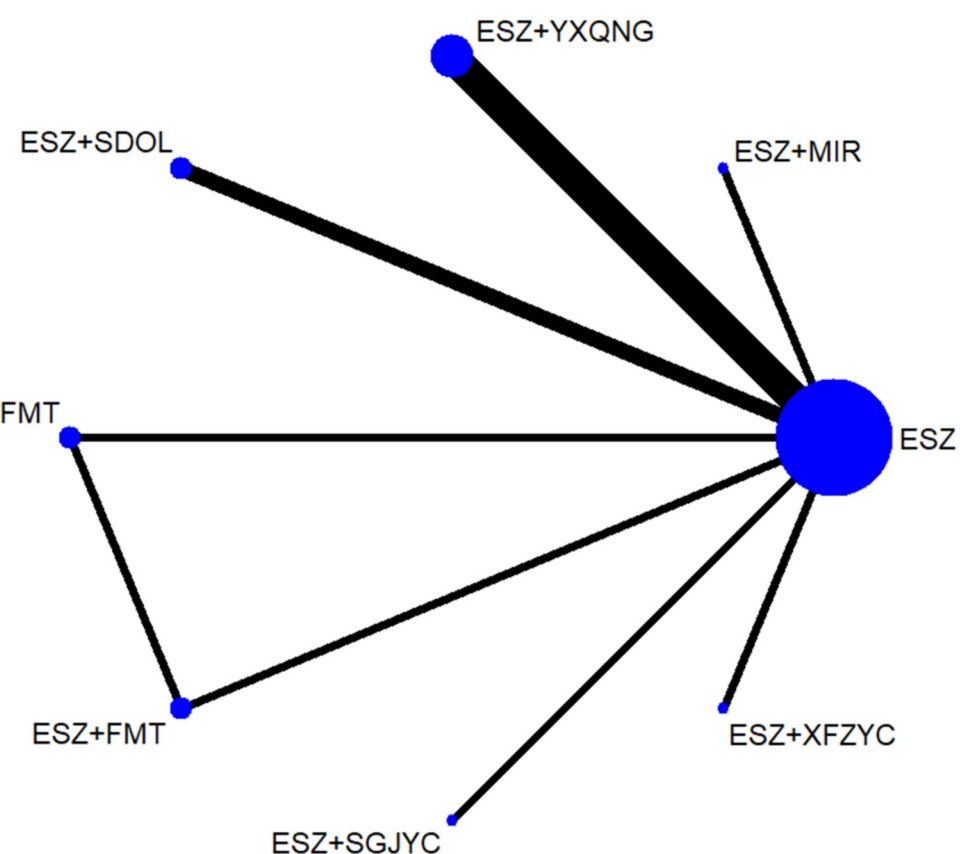

**Fig 5. Network diagram of clinical total effective rate.**

with ESZ+YXONG, the clinical total effective rates of FMT were lower than those of ESZ +YXONG (OR = 0.22, 95% CI = 0.07 to 0.68). Compared with ESZ+SDOL, the clinical total effective rates of FMT were lower than those of ESZ+SDOL (OR = 0.18, 95% CI = 0.04 to 0.85). Compared with FMT, the total clinical effective rates of ESZ+FMT (MD = 4.75, 95% CI = 1.21 to 18.74) and ESZ+XFZYC (MD = 6.82, 95% CI = 1.05 to 44.14) were higher than those of FMT, and there were no significant differences in other comparisons, as shown in Table 4.

**SUCRA.** The specific SUCRA values of each indicator are shown in Table 5.

**Table 4. Network meta-analysis of clinical total effective rate.**

| Treatment | ESZ+XFZYC | ESZ+SGJYC | ESZ+FMT | FMT | ESZ+SDOL | ESZ+YXQNG | ESZ+MIR | ESZ |
|---|---|---|---|---|---|---|---|---|
| ESZ+XFZYC | - | | | | | | | |
| ESZ+SGJYC | 1.11 (0.12,10.30) | - | | | | | | |
| ESZ+FMT | 1.43 (0.18,11.65) | 1.30 (0.16,10.61) | - | | | | | |
| FMT | 6.82 (1.05,44.14)* | 6.17 (0.95,40.23) | 4.75 (1.21,18.74)* | - | | | | |
| ESZ+SDOL | 1.26 (0.18,8.80) | 1.14 (0.16,8.02) | 0.88 (0.15,5.28) | 0.18 (0.04,0.85)* | - | | | |
| ESZ+YXQNG | 1.48 (0.28,7.79) | 1.34 (0.25,7.11) | 1.03 (0.23,4.54) | 0.22 (0.07,0.68)* | 1.17 (0.33,4.17) | - | | |
| ESZ+MIR | 1.10 (0.12,10.38) | 1.00 (0.11,9.44) | 0.77 (0.09,6.35) | 0.16 (0.02,1.07) | 0.88 (0.12,6.26) | 0.75 (0.14,4.03) | - | |
| ESZ | 5.98 (1.24,28.83)* | 5.41 (1.11,26.31)* | 4.17 (1.05,16.61)* | 0.88 (0.32,2.40) | 4.76 (1.51,14.95)* | 4.05 (2.36,6.95)* | 5.43 (1.10,26.83)* | - |

**Table 5. SUCRA.**

| Treatment | SUCRA (%) | | |
|---|---|---|---|
| | PSQI | MESSS | Clinical total effective rate |
| ESZ+SDOL | 99.5 | 93.3 | 63.8 |
| ESZ+SGJYC | 67.3 | - | 66.9 |
| ESZ+AGO | 65.6 | 46.6 | - |
| ESZ+FMT | 63.4 | - | 58.6 |
| ESZ+YXQNG | 54.6 | - | 56.3 |
| ESZ+MIR | 35.1 | - | 67.1 |
| ESZ | 9.2 | 0.1 | 9.6 |
| FMT | 5.3 | - | 7.4 |
| ESZ+XFZYC | - | - | 70.4 |

**PSQI.** According to the results of SUCRA, ESZ+SDOL may be the most effective intervention to reduce PSQI. The ranking results of SUCRA probability are as follows: ESZ+SDOL (99.5%)>ESZ+SGJYC (67.3%)>ESZ+AGO (65.6%)>ESZ+FMT (63.4%)>ESZ+YXQNG (54.6%)>ESZ+MIR (35.1%)>ESZ (9.2%)>FMT (5.3%). The cumulative probability ranking diagram is shown in Fig 6A. The larger the area under the curve, the more effective it is.

**MESSS.** The SUCRA results suggest that ESZ+SDOL may be the most effective intervention measure to increase the clinical total effective rate. As seen in Fig 6B, the ranking results of SUCRA probability are as follows: ESZ+SDOL (93.3%)>ESZ+AGO (46.6%)>ESZ (0.1%).

**Clinical total effective rate.** According to the results of SUCRA, ESZ+XFZYC may be the most effective intervention in the clinic. As seen in Fig 6C, the SUCRA probability ranking results are as follows: ESZ+XFZYC (70.4%)>ESZ+MIR (67.1%)>ESZ+SGJYC (66.9%)>ESZ+SDOL (63.8%)>ESZ+FMT (58.6%)>ESZ+YXQNG (56.3%)>ESZ (9.6%)>FMT (7.4%).

## Security

Of the 18 studies included, 9 [15, 16, 18, 20, 22–24, 26, 29] reported specific information on adverse reactions. The main adverse reactions were nausea, vomiting, abdominal distension, drowsiness, headache, diarrhoea, abdominal pain, dry mouth, tremor, infection, dyspepsia, dizziness, bitter mouth, palpitation, rash dermatitis, gastrointestinal reactions, fatigue, and coordination disorders, as seen in Tables 5 and 6. Among them, 380 patients in the ESZ group had 17 kinds of adverse reactions; 126 patients in the ESZ+SGJYC group had 6 kinds of adverse reactions; 34 patients in the ESZ+SDOL group had no adverse reactions; 30 patients in the ESZ+ESC group had 6 kinds of adverse reactions; 30 patients in the ESC group had 5 kinds of adverse reactions; 150 patients in the ESZ+YXQNG group had 8 kinds of adverse reactions; 38 patients in the FMT group had 7 kinds of adverse reactions; and 38 patients in the ESZ+FMT group had 7 kinds of adverse reactions, as seen in Tables 6 and 7.

## Discussion

Insomnia after stroke is the most common sequela after stroke. It is well known that good sleep quality is one of the benign factors for the prognosis of stroke. In recent years, there have been an increasing number of clinical trials of drug interventions for insomnia after stroke, and eszopiclone has gradually become one of the main drugs in the research of insomnia after stroke. Its advantages are small dependence, fast onset and good clinical efficacy. The literature included in this study involves 11 kinds of drugs combined with ESZ, of which ESZ+SDOL, ESZ+SGJYC, ESZ+YXQNG and ESZ+XFZYC are all Chinese patent medicines combined

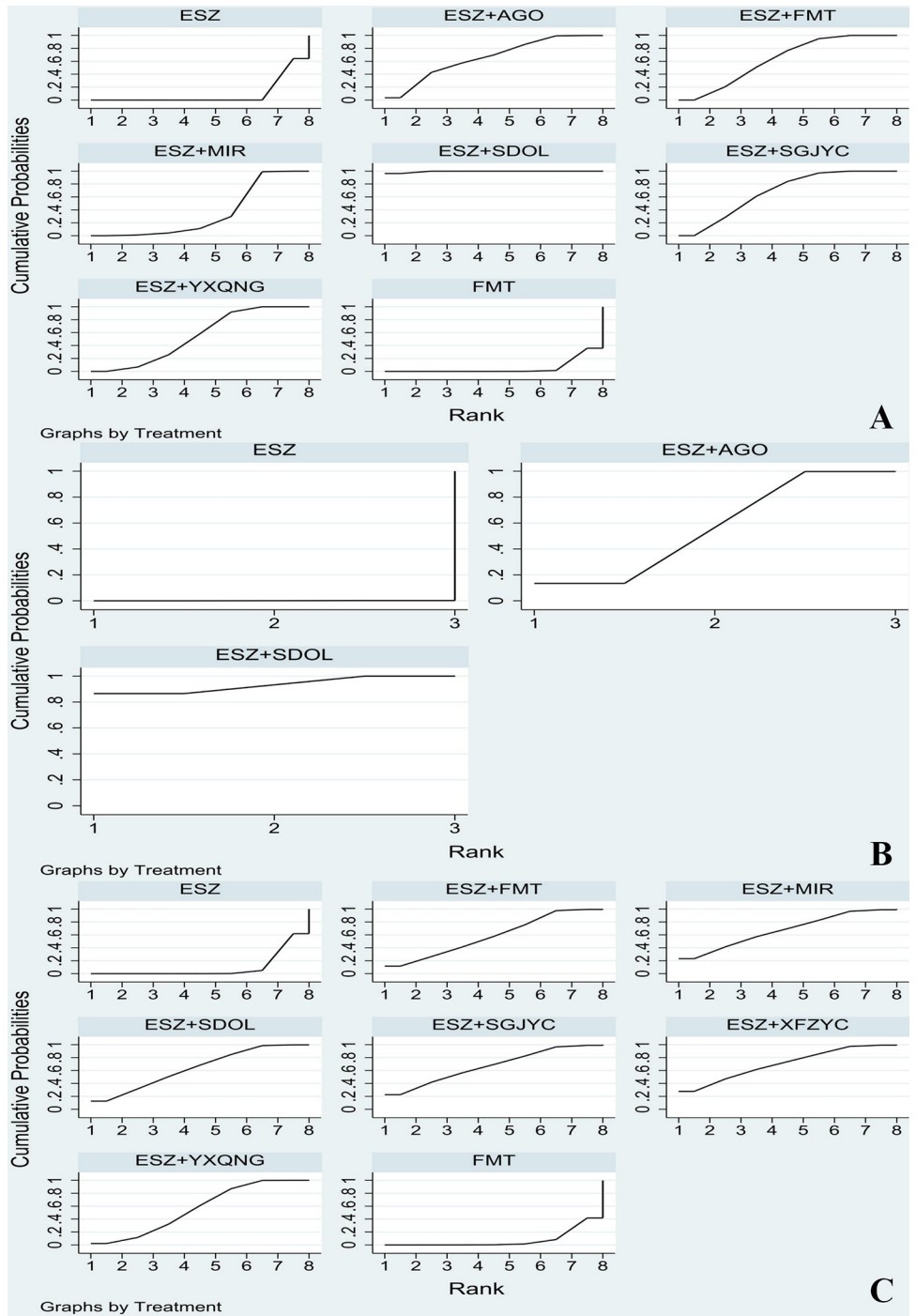

**Fig 6. SUCRA (A: PSQI; B: MESSS; C: Clinical total effective rate).**

with ESZ. The difference lies in the different components and action targets in the patent medicine; ESZ+MIR, ESZ+AGO and ESZ+ESC are combination therapies of ESZ and antidepressants; FMT in ESZ+FMT are a compound preparation, which is composed of flupentixol and melitracen, and it has good clinical efficacy in the treatment of anxiety and depression. Many meta-analyses have suggested the significant clinical efficacy of ESZ in the treatment of

**Table 6. Analysis of adverse reactions (Number of events).**

| Treatment | Total number | nausea | vomit | abdominal distension | drowsiness | headache | diarrhea | abdominal pain | dry mouth | tremor |
|---|---|---|---|---|---|---|---|---|---|---|
| ESZ | 380 | 10 | 6 | 2 | 4 | 7 | 3 | 2 | 4 | 0 |
| ESZ+SGJYC | 126 | 3 | 3 | 2 | 2 | 0 | 2 | 1 | 0 | - |
| ESZ+SDOL | 34 | - | - | - | 0 | 0 | - | - | - | - |
| ESZ+ESC | 30 | 9 | - | - | 7 | - | - | 12 | - | - |
| ESC | 30 | 8 | - | - | 8 | - | - | 11 | - | - |
| ESZ+YXQNG | 150 | 4 | 2 | - | - | 3 | 4 | - | - | - |
| FMT | 38 | 2 | 2 | - | - | - | - | - | 2 | 1 |
| ESZ+FMT | 38 | 2 | 2 | - | - | - | - | - | 3 | 1 |

insomnia [30–32], but the number of meta-analyses related to ESZ and its combination therapy in the treatment of insomnia after stroke is still small. This study combined the results of previous RCTs with the network meta-analysis method to evaluate the clinical efficacy and safety of ESZ combined with drug therapy in the treatment of insomnia after stroke and finally ranked the results according to efficacy.

## Clinical efficacy

The PSQI was developed by Dr. Buysse et al., a psychiatrist at the University of Pittsburgh in the United States in 1989 [33]. The scale is suitable for evaluating sleep quality in patients with sleep disorders and mental disorders, as well as for evaluating sleep quality in the general population. The PSQI is used to evaluate sleep quality and sleep disorders in 1-month intervals. Nineteen individual items generate seven "constituent" scores: subjective sleep quality, sleep latency, and sleep duration, the sum of the scores for habitual sleep efficiency, sleep disorders, use of sleeping medications, and daytime dysfunction yields a global score. The score is directly proportional to the degree of sleep disorder of the subject, meaning that the higher the score is, the more severe the patient's sleep disorder is. In this study, the top four SUCRA ranking results of PSQI were ESZ+SDOL, ESZ+SGJYC, ESZ+AGO, and ESZ+FMT. However, the top four ranking results of clinical total efficacy in this study were ESZ+XFZYC, ESZ+MIR, ESZ+SGJYC, and ESZ+SDOL. Notably, the top four SUCRA values of clinical total efficacy are very close, indicating similar efficacy. Combining two key outcome indicators, in this study, SUCRA suggests that ESZ+SDOL has the best efficacy in improving sleep quality in stroke patients, followed by ESZ+SGJYC and ESZ+XFZYC. Modern pharmacological studies have confirmed that SDOL has sedative and anti-anxiety

**Table 7. Analysis of adverse reactions (Number of events).**

| Treatment | Total number | infection | dyspepsia | dizziness | bitter mouth | palpitation | rash dermatitis | gastrointestinal reactions | fatigue | coordination disorders |
|---|---|---|---|---|---|---|---|---|---|---|
| ESZ | 380 | 1 | 1 | 7 | 8 | 1 | 4 | 3 | 2 | 1 |
| ESZ+SGJYC | 126 | 0 | 0 | 0 | - | - | - | - | - | - |
| ESZ+SDOL | 34 | - | - | - | - | - | - | - | - | - |
| ESZ+ESC | 30 | - | - | 4 | 4 | 4 | - | - | - | - |
| ESC | 30 | - | - | 2 | 0 | 6 | - | - | - | - |
| ESZ+YXQNG | 150 | - | - | 2 | - | - | 1 | 1 | - | 1 |
| FMT | 38 | - | - | 1 | 2 | - | - | - | 1 | - |
| ESZ+FMT | 38 | - | - | 3 | 3 | - | - | - | 2 | - |

effects. The medicine consists of Acanthopanax senticosus and Polygonatum sibiricum, which are sovereign drugs and can strengthen the brain and calm nerves. Rehmannia glutinosa, mulberry, Lycium barbarum, epimedium and male silkworm moth are the official medicines, which affect sex hormones and play the role of nourishing kidney yin and kidney yang. Adjuvants such as Poria cocos, Pericarpium Citri Reticulatae and Pinellia ternata can strengthen the spleen and stomach, promote qi and relieve depression, promote gastrointestinal digestive function and increase appetite. The medicinal nux vomica has the effect of dredging meridians, and a small amount of nux vomica has an excitatory effect on the cerebral cortex and spinal cord. SDOL can not only help sleep and calm the mind but also relieve the negative emotions of patients with long-term insomnia, effectively improving memory and protecting brain cell function [34, 35]. The main components of SGJYC are Hypericum perforatum and Acanthopanax senticosus. However, in relevant pharmacological studies [36–38], it has been suggested that it can improve depression and insomnia in patients by regulating neurotransmitters and improving the synaptic plasticity of neurons. Effective insomnia and depression are closely related at the symptomatology and disease levels. A total of 40%-92% of insomnia symptoms are caused by mental diseases [39, 40]. Studies have shown that a shugan jieyu capsule combined with antidepressants can effectively improve the PSQI index of depressed patients with sleep disorders [41]. The prototype of XFZYC is xuefu zhuyu Decoction (XFZYD), a classic traditional Chinese medicine prescription based on the clinical efficacy of activating blood circulation and removing blood stasis. Its components can significantly improve blood supply, promote blood circulation, and control the recurrence of thrombosis. Studies have confirmed that XFZYD has neuroprotective effects against brain injury caused by ischaemic stroke through the lymphatic system [42], but there are few studies on its clinical efficacy in insomnia after stroke. This may also be one of the future research priorities regarding the drug.

MESSS, another secondary index in this study, is a commonly used scoring system in the clinical neuropathy score, which is used to describe the severity of nerve injury clinically. In this study, the SUCRA ranking results of messages were ESZ+SDOL, ESZ+AGO, and ESZ. The results showed that ESZ+SDOL was the best intervention to improve the nerve injury of patients. Some studies on the effect of SDOL intervention on sleep deprivation model rats found that the spatial learning and memory ability of rats in the SDOL group was increased, the escape latency was shortened, the serum IL-1β and TNF-α content decreased, the content of 5-HT in the raphe nucleus decreased, and the BDNF protein level in the hippocampus was significantly upregulated [43], which may also be one of the mechanisms by which SDOL improves insomnia and nerve injury in patients after stroke.

## Security

Because the specific adverse reactions in the individual included studies were vague, we only included the number of patients with specific adverse reactions (such as nausea, vomiting, dizziness, etc.). The control group included in this study was ESZ therapy, so it was not compared with other therapies. The results of this study showed that in addition to ESZ therapy, the number of adverse reactions of ESZ+ESC was the greatest, and abdominal pain was the greatest, while the types of adverse reactions of ESZ+YXQNG were the greatest, including nausea, vomiting, headache, diarrhoea, dizziness, rash dermatitis, gastrointestinal reactions and coordination disorders. Studies including ESZ+SDOL, which had the best clinical efficacy in this study, did not mention adverse reactions. In addition to the above therapies, other drug therapies have 5–7 kinds of adverse reactions. A total of 9 included studies in this study mentioned adverse reactions after intervention with ESZ combined with other drugs, but there was no

direct evidence to confirm the direct connection between adverse reactions and drugs. At the same time, due to the small number of studies and the differences in the course of treatment or other factors in the literature, it is only for reference.

## Limitations

To date, this is the first meta-analysis of ESZ combined with other drugs in the treatment of insomnia after stroke. However, this study has the following limitations: 1) This study included only RCTs related to the intervention of ESZ-based drug combination therapy on insomnia after stroke and did not include RCTs related to the intervention of other drugs on insomnia after stroke, mainly because there were too many studies related to the intervention of other drugs on insomnia after stroke during the previous search of relevant literature, which may require a huge workload. In the future, we will consider screening high-quality RCTs to make a network comparison of different drugs in the intervention of insomnia after stroke. 2. Most of the literature included in this study was of poor quality, and most RCTs of distributive concealment, blinding and potential bias were not mentioned. 3. The baseline included in the study may increase the possibility of inconsistency and clinical heterogeneity if the course of treatment is different.

## Conclusion

In summary, multi-drug therapy based on ESZ can improve the sleep quality of insomnia patients after stroke to varying degrees. Current evidence suggests that ESZ+SDOL and ESZ+SGJYC may have the best effect in reducing PSQI; In terms of reducing MESSS, ESZ+SDOL may have the best effect; In terms of clinical efficacy in improving sleep, ESZ+XFZYC has the best clinical efficacy. Further quantitative analysis is needed for adverse reactions. Due to the limitations of the quantity and quality of included literature, the above conclusions still need to be verified by more high-quality RCTs.

## Supporting information

**S1 Checklist. PRISMA NMA checklist of Items to include when reporting a systematic review involving a network meta-analysis.**
(DOCX)

**S1 Appendix. Search strategy.**
(DOCX)

**S2 Appendix. Instruction code for meta.**
(DOC)

**S1 Table. Basic characteristics of trials included (supplementary).**
(DOC)

**S1 Data. Raw data.**
(XLSX)

## Author Contributions

**Data curation:** De-Liang Zhu.

**Formal analysis:** De-Liang Zhu.

**Software:** Ke-Yu Chen.

**Supervision:** Ke-Yu Chen.

**Validation:** Ke-Yu Chen.

**Writing – original draft:** Ruo-Yang Li.

**Writing – review & editing:** Ruo-Yang Li.

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
