## [Decision Letter · Decision Letter 0]

16 Oct 2023

PONE-D-23-27985Efficacy and safety of eszopiclone combined with drug therapy in the treatment of insomnia after stroke: A network meta-analysis and systematic reviewPLOS ONE

Dear Dr. Li,

Thank you for submitting your manuscript to PLOS ONE. After careful consideration, we feel that it has merit but does not fully meet PLOS ONE’s publication criteria as it currently stands. Therefore, we invite you to submit a revised version of the manuscript that addresses the points raised during the review process.

ACADEMIC EDITOR: Kindly ensure that you address all the comments raised by the reviewers and conduct a thorough proofreading of the article to rectify any grammatical and English errors.

We look forward to receiving your revised manuscript.

Kind regards,

Sidhant Ochani, MBBS

Academic Editor

PLOS ONE

Journal Requirements:

4. Kindly register your study protocol for network meta-analysis.

Additional Editor Comments:

Dear Authors,

I hope this message finds you well. Following a thorough review of the manuscript, it has come to our attention that there are several areas requiring revision and justification, as highlighted by the comments provided by the reviewers. To enhance the overall quality of your manuscript, we recommend seeking assistance from experts in Network Meta Analysis. Additionally, it would be beneficial to refer to the AMSTAR checklist to assess and ensure the robustness of your research.

Your attention to these aspects will significantly contribute to the refinement of the manuscript, aligning it more closely with the standards expected for publication.

Reviewers' comments:

Reviewer's Responses to Questions

**Comments to the Author**

1. Is the manuscript technically sound, and do the data support the conclusions?

Reviewer #1: Yes

Reviewer #2: No

2. Has the statistical analysis been performed appropriately and rigorously? 

Reviewer #1: I Don't Know

Reviewer #2: No

3. Have the authors made all data underlying the findings in their manuscript fully available?

Reviewer #1: Yes

Reviewer #2: No

4. Is the manuscript presented in an intelligible fashion and written in standard English?

Reviewer #1: No

Reviewer #2: No

5. Review Comments to the Author

Reviewer #1: This paper underscores the importance of addressing insomnia in stroke patients and highlights the concerns surrounding the use of benzodiazepines for this purpose. Instead, it introduces eszopiclone (ESZ), a newer nonbenzodiazepine sedative, as a potential treatment option, given its favorable pharmacokinetics for stroke patients. The significance of this paper lies in its investigation of the safety and efficacy of ESZ when combined with Chinese medications for post-stroke insomnia treatment. It emphasizes the need for a systematic analysis of this condition, which has received limited attention compared to other stroke-related issues. Employing a network meta-analysis approach with outcome indicators like the Pittsburgh Sleep Quality Index (PSQI), this research can provide valuable insights for clinical practitioners, aiding them in informed decision-making for post-stroke insomnia treatment and potentially improving patient outcomes.

However, it's essential to acknowledge certain limitations of this study. Firstly, it exclusively focuses on randomized controlled trials (RCTs) involving ESZ-based drug combinations and excludes RCTs exploring alternative drug interventions due to the extensive existing literature. Secondly, many of the included studies lack essential details regarding concealment, blinding, and potential bias, which affects the overall reliability of the findings. Lastly, variations in treatment duration and baseline characteristics among the studies may introduce inconsistencies and clinical heterogeneity into the analysis, potentially impacting result validity. Nevertheless, this study marks a crucial step forward in understanding post-stroke insomnia treatment strategies.

However, there are several significant issues that need to be addressed for the manuscript to progress.

**Major Issues:**

1. It is essential to provide an explanation for the choice of keywords, particularly the omission of "nonbenzodiazepine" and similar mesh terms. The current set of keywords does not appear to result in a specific and comprehensive literature search.

2. Additionally, please include reasons for excluding specific types of original studies, such as observational cohorts, in order to justify the choices made.

2. The manuscript should offer clear reasoning for opting for a fixed model when there was heterogeneity among studies, especially regarding specific treatments. Justification for this choice is crucial to ensure the robustness of the analysis.

**Minor Changes:**

1. It is advisable to include a conclusion.

2. Figures 3, 4B, and 6 are currently of low resolution and readability.

3. Table captions should undergo revision to correct spelling errors. Additionally, in Table 1, please adjust the formatting for the entry "Song YM [19]" to ensure consistency.

4. The keywords should be revised to eliminate repetitions and enhance clarity.

Reviewer #2: I appreciate the authors' efforts in conducting this study; however, there are several critical concerns that need to be addressed before the manuscript can be considered for publication. Below are my detailed comments:

Abstract:

The use of uncommon and undefined abbreviations should be avoided in the abstract. The non-standard abbreviations used in the results section reduce readability and conciseness.

Introduction:

2. In line 6, the statement "there is still limited systematic analysis on poststroke insomnia" should be more specific. It would be more informative to highlight the shortcomings of previous meta-analyses on post-stroke insomnia rather than simply stating it as "limited."

Methods:

(1) The phrase "free words" is not commonly used; "keywords" is a more appropriate term.

(2) It is unclear how the search terms were adjusted based on the search results. Was there a standardized process, or was it done subjectively?

(3) The statement "If it was not calculated in the original text, it was calculated on its own" needs clarification. What does "calculated on its own" mean?

(4) The statistical methodology lacks detail and reproducibility. Please specify whether a frequentist or Bayesian network meta-analysis approach was employed, and provide information on the software packages or commands used.

Results:

(1) In the "Literature search results" section, it is mentioned that traditional Chinese medicine such as "shugan jieyu capsule (ESZ+SGJYC)" was used, but only the Pinyin transliteration is provided. This may not be clear to many readers and should be explained.

(2) Table 1 is missing essential patient characteristics such as disease duration and nationality.

(3) The quality of figures is subpar with blurry text, poor screenshot quality, and messy formatting. It does not meet academic standards.

(4) The results section lacks specific numerical values and significance comparisons between therapies. The descriptions are mainly qualitative, and the numbers in the figures are barely readable. The paper needs to present numerical data for meaningful interpretation. Additionally, SUCRA values should be reported as numerical values, not just curves. The authors' network meta-analysis appears superficial, and they seem to lack familiarity with the standards for figure creation and academic writing in this field.

Conclusion:

5. The conclusion is too shallow and does not reflect the true clinical value and significance of the meta-analysis. It does not serve as a guidance for evidence-based medicine.

Other:

The English language used in the manuscript requires improvement. Expressions like "combined drug intervention therapy" and "clinical workers" sound unusual.

There are minor spelling errors, such as "Literature inclusion criteria" in the fourth line, which should be corrected.

I recommend that the authors carefully address these issues to enhance the manuscript's quality and scientific rigor.

6. PLOS authors have the option to publish the peer review history of their article (what does this mean?). If published, this will include your full peer review and any attached files.

Reviewer #1: **Yes: **Amna Siddiqui

Reviewer #2: No

---

## [Author Response · Author response to Decision Letter 0]

24 Oct 2023

Dear editor and reviewers

1.Please ensure that your manuscript meets PLOS ONE's style requirements, including those for file naming.

Reply: I have adjusted the format of the document according to the requirements.

2.Note from Emily Chenette, Editor in Chief of PLOS ONE, and Iain Hrynaszkiewicz, Director of Open Research Solutions at PLOS: Did you know that depositing data in a repository is associated with up to a 25% citation advantage (https://doi.org/10.1371/journal.pone.0230416)? If you’ve not already done so, consider depositing your raw data in a repository to ensure your work is read, appreciated and cited by the largest possible audience. You’ll also earn an Accessible Data icon on your published paper if you deposit your data in any participating repository 

Reply: I have uploaded the data as a supporting information file.

3.Please include captions for your Supporting Information files at the end of your manuscript, and update any in-text citations to match accordingly. Please see our Supporting Information guidelines for more information:

Reply: I have referenced the supporting information file according to your request.

4.Kindly register your study protocol for network meta-analysis.

Reply: I have registered according to your requirements, and the registration number has been added in the first chapter of the method.

Additional Editor Comments:

I hope this message finds you well. Following a thorough review of the manuscript, it has come to our attention that there are several areas requiring revision and justification, as highlighted by the comments provided by the reviewers. To enhance the overall quality of your manuscript, we recommend seeking assistance from experts in Network Meta Analysis. Additionally, it would be beneficial to refer to the AMSTAR checklist to assess and ensure the robustness of your research.

Your attention to these aspects will significantly contribute to the refinement of the manuscript, aligning it more closely with the standards expected for publication.

Reply: Thank you very much for your suggestion. I have made certain modifications according to the expert's requirements and also evaluated my research using the AMSTAR checklist.

Reviewer #1: This paper underscores the importance of addressing insomnia in stroke patients and highlights the concerns surrounding the use of benzodiazepines for this purpose. Instead, it introduces eszopiclone (ESZ), a newer nonbenzodiazepine sedative, as a potential treatment option, given its favorable pharmacokinetics for stroke patients. The significance of this paper lies in its investigation of the safety and efficacy of ESZ when combined with Chinese medications for post-stroke insomnia treatment. It emphasizes the need for a systematic analysis of this condition, which has received limited attention compared to other stroke-related issues. Employing a network meta-analysis approach with outcome indicators like the Pittsburgh Sleep Quality Index (PSQI), this research can provide valuable insights for clinical practitioners, aiding them in informed decision-making for post-stroke insomnia treatment and potentially improving patient outcomes.

However, it's essential to acknowledge certain limitations of this study. Firstly, it exclusively focuses on randomized controlled trials (RCTs) involving ESZ-based drug combinations and excludes RCTs exploring alternative drug interventions due to the extensive existing literature. Secondly, many of the included studies lack essential details regarding concealment, blinding, and potential bias, which affects the overall reliability of the findings. Lastly, variations in treatment duration and baseline characteristics among the studies may introduce inconsistencies and clinical heterogeneity into the analysis, potentially impacting result validity. Nevertheless, this study marks a crucial step forward in understanding post-stroke insomnia treatment strategies.

However, there are several significant issues that need to be addressed for the manuscript to progress.

**Major Issues:**

1. It is essential to provide an explanation for the choice of keywords, particularly the omission of "nonbenzodiazepine" and similar mesh terms. The current set of keywords does not appear to result in a specific and comprehensive literature search.

Reply: Your suggestion has been very helpful in adjusting my search strategy. I have revised the entire search strategy according to your requirements, as detailed in Appendix S1.

2. Additionally, please include reasons for excluding specific types of original studies, such as observational cohorts, in order to justify the choices made.

Reply: Thank you for your advise，The meta-analysis of comprehensive high-quality randomized controlled trials has been regarded as the highest level of evidence in evidence-based medicine. Therefore, this study only includes RCTs and excludes all other types of studies, as there are not too few included studies.I also added the reason to the exclusion section of the study.

3. The manuscript should offer clear reasoning for opting for a fixed model when there was heterogeneity among studies, especially regarding specific treatments. Justification for this choice is crucial to ensure the robustness of the analysis.

Reply: Thank you very much for correcting my mistake in choosing the model. I deeply understand that choosing a fixed model when there is heterogeneity is not appropriate, and have adjusted it to a random effects model.

**Minor Changes:**

1. It is advisable to include a conclusion.

Reply: The conclusion has been supplemented and modified according to your requirements.

2. Figures 3, 4B, and 6 are currently of low resolution and readability.

Reply: I have made corresponding modifications to the image quality in the article according to your requirements.

3. Table captions should undergo revision to correct spelling errors. Additionally, in Table 1, please adjust the formatting for the entry "Song YM [19]" to ensure consistency.

Reply: Spelling errors have been corrected and formatting has been adjusted.

4. The keywords should be revised to eliminate repetitions and enhance clarity.

Reply: I have made corresponding modifications to some keywords according to your requirements.

Reviewer #2: I appreciate the authors' efforts in conducting this study; however, there are several critical concerns that need to be addressed before the manuscript can be considered for publication. Below are my detailed comments:

Abstract:

The use of uncommon and undefined abbreviations should be avoided in the abstract. The non-standard abbreviations used in the results section reduce readability and conciseness.

Reply: I have defined the uncommon abbreviations in the abstract based on your suggestion.

Introduction:

2. In line 6, the statement "there is still limited systematic analysis on poststroke insomnia" should be more specific. It would be more informative to highlight the shortcomings of previous meta-analyses on post-stroke insomnia rather than simply stating it as "limited."

Reply: Your suggestion is very useful for my research, and I have made corresponding additions and modifications to this sentence based on your suggestion.

Methods:

(1) The phrase "free words" is not commonly used; "keywords" is a more appropriate term.

Reply: I have made corresponding modifications based on your suggestion.

(2) It is unclear how the search terms were adjusted based on the search results. Was there a standardized process, or was it done subjectively?

Reply: The search term is RCT, stroke, Insomnia, and Eszopiclone as the theme words, and their keywords are searched in the search library. Synonyms are connected by or, and different words are connected by and. Please refer to the appendix of S1 for specific search libraries.

(3) The statement "If it was not calculated in the original text, it was calculated on its own" needs clarification. What does "calculated on its own" mean?

Reply: I have added the self calculated formula to the manuscript.

(4) The statistical methodology lacks detail and reproducibility. Please specify whether a frequentist or Bayesian network meta-analysis approach was employed, and provide information on the software packages or commands used.

Reply: I have supplemented this section based on your suggestion and uploaded the software commands as an S2 appendix.

Results:

(1) In the "Literature search results" section, it is mentioned that traditional Chinese medicine such as "shugan jieyu capsule (ESZ+SGJYC)" was used, but only the Pinyin transliteration is provided. This may not be clear to many readers and should be explained.

Reply: Thank you very much for your insightful suggestion. We have provided a simple explanation for it.

(2) Table 1 is missing essential patient characteristics such as disease duration and nationality.

Reply: Thank you very much for your suggestion. Due to the limited space of the table, I have uploaded it as S1 table format.

(3) The quality of figures is subpar with blurry text, poor screenshot quality, and messy formatting. It does not meet academic standards.

Reply: I have made corresponding modifications to the image quality in the article according to your requirements.

(4) The results section lacks specific numerical values and significance comparisons between therapies. The descriptions are mainly qualitative, and the numbers in the figures are barely readable. The paper needs to present numerical data for meaningful interpretation. Additionally, SUCRA values should be reported as numerical values, not just curves. The authors' network meta-analysis appears superficial, and they seem to lack familiarity with the standards for figure creation and academic writing in this field.

Reply: The conclusion has been supplemented and modified according to your requirements.

Conclusion:

5. The conclusion is too shallow and does not reflect the true clinical value and significance of the meta-analysis. It does not serve as a guidance for evidence-based medicine.

Reply: The conclusion has been supplemented and modified according to your requirements.

Other:

The English language used in the manuscript requires improvement. Expressions like "combined drug intervention therapy" and "clinical workers" sound unusual.

There are minor spelling errors, such as "Literature inclusion criteria" in the fourth line, which should be corrected.

Reply: Thank you very much for your suggestion. I have made corresponding corrections to the English language and spelling issues in the manuscript.

---

## [Decision Letter · Decision Letter 1]

5 Dec 2023

PONE-D-23-27985R1Efficacy and safety of eszopiclone combined with drug therapy in the treatment of insomnia after stroke: A network meta-analysis and systematic reviewPLOS ONE

Dear Dr. Li,

Thank you for submitting your manuscript to PLOS ONE. After careful consideration, we feel that it has merit but does not fully meet PLOS ONE’s publication criteria as it currently stands. Therefore, we invite you to submit a revised version of the manuscript that addresses the points raised during the review process.

We look forward to receiving your revised manuscript.

Kind regards,

Huijuan Cao, Ph.D.

Academic Editor

PLOS ONE

Journal Requirements:

Reviewers' comments:

Reviewer's Responses to Questions

**Comments to the Author**

1. If the authors have adequately addressed your comments raised in a previous round of review and you feel that this manuscript is now acceptable for publication, you may indicate that here to bypass the “Comments to the Author” section, enter your conflict of interest statement in the “Confidential to Editor” section, and submit your "Accept" recommendation.

Reviewer #1: All comments have been addressed

Reviewer #3: All comments have been addressed

Reviewer #4: All comments have been addressed

2. Is the manuscript technically sound, and do the data support the conclusions?

Reviewer #1: Yes

Reviewer #3: (No Response)

Reviewer #4: Yes

3. Has the statistical analysis been performed appropriately and rigorously? 

Reviewer #1: I Don't Know

Reviewer #3: (No Response)

Reviewer #4: Yes

4. Have the authors made all data underlying the findings in their manuscript fully available?

Reviewer #1: Yes

Reviewer #3: (No Response)

Reviewer #4: Yes

5. Is the manuscript presented in an intelligible fashion and written in standard English?

Reviewer #1: Yes

Reviewer #3: (No Response)

Reviewer #4: Yes

6. Review Comments to the Author

Reviewer #1: (No Response)

Reviewer #3: (No Response)

Reviewer #4: Comments to the Submitting Author,

The authors evaluated the efficacy and safety of multi-drug therapy based on eszopiclone in the treatment of insomnia after stroke based on a network meta-analysis method. The results of this NMA provide some ideas for clinicians in practical applications and also provide some evidence for relevant decision makers in the development of guidelines. Currently, based on the previous peer reviewer's comments, the modifications made by the authors are acceptable. However, before formal publication, there are still some minor issues regarding citation of literature and resolution of images.

1. I am not sure if the authors of the 'Data extraction' chapter used the Cochrane Handbook as the source for their formulas, but it seems appropriate to cite the source of the relevant formulas.The formula is currently described in the latest version of the Handbook (https://training.cochrane.org/handbook/current/chapter-06#section-6-5-2-8):

Plz cite this chapter as: Higgins JPT, Li T, Deeks JJ (editors). Chapter 6: Choosing effect measures and computing estimates of effect. In: Higgins JPT, Thomas J, Chandler J, Cumpston M, Li T, Page MJ, Welch VA (editors). Cochrane Handbook for Systematic Reviews of Interventions version 6.4 (updated August 2023). Cochrane, 2023. Available from www.training.cochrane.org/handbook.

2. At present, the RoB tool used by the author has certain problems, such as not considering special types of RCTs such as cluster randomization and crossover design, not considering the balance of inter group baselines, not clearly defining the effects of intervention allocation and compliance, and missing inter group contamination. Therefore, Cochrane launched a new version of the RCT Risk of Bias Assessment Tool (RoB2) in 2016, which became more comprehensive on the basis of RoB1. After multiple revisions, RoB2 has now become a widely applicable risk of bias assessment tool (https://training.cochrane.org/handbook/current/chapter-08#section-8-1). I hope that researchers can refer to the research methods introduced in the latest version of Cochrane Handbook to implement their own research in future related studies. In addition, please supplement the references of Cochrane Handbook in the 'Risk assessment' section.

3. Regarding the network plot in this article, the styles of the images uploaded by the authors seem to be different. Fig 5 has a background and background color compared to the other two.

In addition, can the authors optimize the layout of the images? Fig 3B and Fig 4B look very strange.

Finally, it is recommended that the author does not adjust the ratio of the length and width of the image, as both Fig 3 and Fig 4 seem to have this issue.

7. PLOS authors have the option to publish the peer review history of their article (what does this mean?). If published, this will include your full peer review and any attached files.

Reviewer #1: **Yes: **Amna Siddiqui

Reviewer #3: **Yes: **Fabrizio d ascenzo

Reviewer #4: **Yes: **Yi Yuan

---

## [Author Response · Author response to Decision Letter 1]

7 Dec 2023

Dear editor and reviewers

Journal Requirements:

Reply: I have checked all my references and ensured that they have not been withdrawn.

Reviewer #4: Comments to the Submitting Author,

The authors evaluated the efficacy and safety of multi-drug therapy based on eszopiclone in the treatment of insomnia after stroke based on a network meta-analysis method. The results of this NMA provide some ideas for clinicians in practical applications and also provide some evidence for relevant decision makers in the development of guidelines. Currently, based on the previous peer reviewer's comments, the modifications made by the authors are acceptable. However, before formal publication, there are still some minor issues regarding citation of literature and resolution of images.

1.I am not sure if the authors of the 'Data extraction' chapter used the Cochrane Handbook as the source for their formulas, but it seems appropriate to cite the source of the relevant formulas.The formula is currently described in the latest version of the Handbook (https://training.cochrane.org/handbook/current/chapter-06#section-6-5-2-8):Plz cite this chapter as: Higgins JPT, Li T, Deeks JJ (editors). Chapter 6: Choosing effect measures and computing estimates of effect. In: Higgins JPT, Thomas J, Chandler J, Cumpston M, Li T, Page MJ, Welch VA (editors). Cochrane Handbook for Systematic Reviews of Interventions version 6.4 (updated August 2023). Cochrane, 2023. Available from https://www.training.cochrane.org/handbook.

Reply: Thank you very much for your suggestion. I have already cited the references for the formula.

2.At present, the RoB tool used by the author has certain problems, such as not considering special types of RCTs such as cluster randomization and crossover design, not considering the balance of inter group baselines, not clearly defining the effects of intervention allocation and compliance, and missing inter group contamination. Therefore, Cochrane launched a new version of the RCT Risk of Bias Assessment Tool (RoB2) in 2016, which became more comprehensive on the basis of RoB1. After multiple revisions, RoB2 has now become a widely applicable risk of bias assessment tool (https://training.cochrane.org/handbook/current/chapter-08#section-8-1). I hope that researchers can refer to the research methods introduced in the latest version of Cochrane Handbook to implement their own research in future related studies. In addition, please supplement the references of Cochrane Handbook in the 'Risk assessment' section.

Reply: Your suggestions have been of great help to me, and I will refer to the research methods introduced in the latest version of the Cochrane Handbook in all my future research and i have supplemented the references of Cochrane Handbook in the 'Risk assessment' section

3. Regarding the network plot in this article, the styles of the images uploaded by the authors seem to be different. Fig 5 has a background and background color compared to the other two.

In addition, can the authors optimize the layout of the images? Fig 3B and Fig 4B look very strange.

Finally, it is recommended that the author does not adjust the ratio of the length and width of the image, as both Fig 3 and Fig 4 seem to have this issue.

Reply: I have made adjustments to the background and layout of Figure 3-5 based on your feedback.

---

## [Decision Letter · Decision Letter 2]

28 Dec 2023

Efficacy and safety of eszopiclone combined with drug therapy in the treatment of insomnia after stroke: A network meta-analysis and systematic review

PONE-D-23-27985R2

Dear Dr. Li,

We’re pleased to inform you that your manuscript has been judged scientifically suitable for publication and will be formally accepted for publication once it meets all outstanding technical requirements.

Kind regards,

Huijuan Cao, Ph.D.

Academic Editor

PLOS ONE

Additional Editor Comments (optional):

Reviewers' comments:

Reviewer's Responses to Questions

**Comments to the Author**

1. If the authors have adequately addressed your comments raised in a previous round of review and you feel that this manuscript is now acceptable for publication, you may indicate that here to bypass the “Comments to the Author” section, enter your conflict of interest statement in the “Confidential to Editor” section, and submit your "Accept" recommendation.

Reviewer #4: All comments have been addressed

2. Is the manuscript technically sound, and do the data support the conclusions?

Reviewer #4: Yes

3. Has the statistical analysis been performed appropriately and rigorously? 

Reviewer #4: Yes

4. Have the authors made all data underlying the findings in their manuscript fully available?

Reviewer #4: Yes

5. Is the manuscript presented in an intelligible fashion and written in standard English?

Reviewer #4: Yes

6. Review Comments to the Author

Reviewer #4: The authors have completed all the modifications based on the peer review comments, and these modifications are acceptable. Manuscript ID PONE-D-23-27985R2 entitled "Efficacy and safety of eszopiclone combined with drug therapy in the treatment of insomnia after stroke: A network meta-analysis and systematic review" can be accepted for publication. Thank you.

7. PLOS authors have the option to publish the peer review history of their article (what does this mean?). If published, this will include your full peer review and any attached files.

Reviewer #4: **Yes: **Yi Yuan

---

## [Editor Report · Acceptance letter]

24 Jan 2024

PONE-D-23-27985R2 

PLOS ONE

Dear Dr. Li, 

I'm pleased to inform you that your manuscript has been deemed suitable for publication in PLOS ONE. Congratulations! Your manuscript is now being handed over to our production team.

Kind regards, 

on behalf of

Dr. Huijuan Cao 

Academic Editor

PLOS ONE